# Autumn shifts in cold tolerance metabolites in overwintering adult mountain pine beetles

**Kirsten M. Thompson** *, **Dezene P. W. Huber**, **Brent W. Murray**

Natural Resources and Environmental Studies, University of Northern British Columbia, Prince George, British Columbia, Canada

☯ These authors contributed equally to this work.
* kthompso@unbc.ca

**Data Availability Statement:** All relevant data are within the paper and its Supporting Information files.

**Funding:** This research was supported by a grant from the Natural Science and Engineering

## Abstract

The mountain pine beetle, *Dendroctonus ponderosae* (Coleoptera: Curculionidae) is a major forest pest of pines in western North America. Beetles typically undergo a one-year life cycle with larval cold hardening in preparation for overwintering. Two-year life cycle beetles have been observed but not closely studied. This study tracks cold-hardening and preparation for overwintering by adult mountain pine beetles in their natal galleries. Adults were collected *in situ* between September and December 2016 for a total of nine time points during 91 days. Concentrations of 41 metabolites in these pooled samples were assessed using quantitative nuclear magnetic resonance (NMR). Levels of glycerol and proline increased significantly with lowering temperature during the autumn. Newly eclosed mountain pine beetles appear to prepare for winter by generating the same cold-tolerance compounds found in other insect larvae including mountain pine beetle, but high on-site mortality suggested that two-year life cycle adults have a less efficacious acclimation process. This is the first documentation of cold acclimation metabolite production in overwintering new adult beetles and is evidence of physiological plasticity that would allow evolution by natural selection of alternate life cycles (shortened or lengthened) under a changing climate or during expansion into new geoclimatic areas.

## Introduction

The mountain pine beetle, *Dendroctonus ponderosae* (Coleoptera: Curculionidae), is an irruptive forest insect native to western North America [1]. In the past 20 years, mountain pine beetle outbreaks of unusual size killed much of the mature pine in British Columbia [2] and expanded beyond their historical range over the Rocky Mountains to Alberta and into the north of British Columbia [3,4]. Cold winters–greater than two weeks at –40°C–are thought to have limited the scope of previous outbreaks by killing off mountain pine beetle brood by freezing [5,6]. Climate change has led to a warming trend in the past 30 years, reducing the length and frequency of reaching and sustaining this temperature threshold [7]. This trend is particularly the case in the early- and late-winter season when overwintering insects would

Research Council of Canada (grant no. NET GP 434810-12) <http://www.nserc-crsng.gc.ca/index_eng.asp>) to the TRIA Network and its research partners (KMT, DPWH, BWM); a Research Project Award <https://www.unbc.ca/graduate-programs/awards/awards-and-scholarships-introduction> grant from the University of Northern British Columbia (KMT); a Natural Science and Engineering Research Council of Canada <http://www.nserc-crsng.gc.ca/index_eng.asp> Discovery Grant (DPWH), the Canada Research Chair program <http://www.chairs-chaires.gc.ca/home-accueil-eng.aspx> (DPWH); The Canada Foundation for Innovation < https://www.innovation.ca/> (DPWH) and the BC Knowledge Development Fund < https://www2.gov.bc.ca/gov/content/governments/about-the-bc-government/technology-innovation/bckdf> (DPWH). The funders had no role in study design, data collection and analysis, decision to publish, or preparation of the manuscript.

**Competing interests:** The authors have declared that no competing interests exist.

normally be the most vulnerable due to a lack of overwintering metabolites, therefore increasing mountain pine beetle winter survival rates [8,9].

Most mountain pine beetles have a univoltine lifecycle, meaning they mature over the course of a single year [10,11]. Life cycle development is dictated by climatic factors; adults lay eggs in the late summer to early fall, allowing for partial larval development prior to freezing winter temperatures. Mountain pine beetles are freeze intolerant and will experience mortality if water within their soft tissue crystalizes [12]. For this reason, mountain pine beetles generate cryoprotectants in the autumn, especially glycerol, to reduce their super cooling point and protect against the formation of ice crystals [13–16]. Early instar larvae (instars 1 and 2) are marginally less cold tolerant than late-instars (instars 3 and 4), with the first three instars generating similar proportions of glycerol in relation to bodyweight, and the final instar producing slightly more [6,17]. Larvae also void their guts in preparation for cooler temperatures in order to reduce the number of internal ice nucleation surfaces [18]. Pupation and maturation follow in the spring and beetles fly in the summer to find new hosts and repeat the life cycle.

Mountain pine beetle phenology is known to be most responsive to temperature changes, with hormonal regulation and photo period playing little part in cold acclimation [13]. Photoperiod, historically thought to have little effect on cold hardening, may be involved to some degree as new evidence suggests that mountain pine beetle larvae respond negatively to light, even when located in a sub-cortical environment [19]. Reliance on temperature occasionally results in phenological delays causing an extension of the life cycle beyond one year. Such larvae overwinter, pupate later in the summer, and overwinter a second time as new adults [20]. Larval cold hardening and cryoprotectant generation in insects is mainly understood [21], with mountain pine beetle-specific studies of cryoprotectant production [13], RNA transcript generation [15,16], and proteomic data [22] suggesting some of the mechanisms and pathways of larvae winter survival. While overwintering new adults are occasionally recorded in the literature [20,23], and have been observed as an early-May flight during the height of the outbreak in central BC (DPWH, personal communication), they represent a less common life strategy and the precise mechanisms of overwintering are unknown. In this study we recorded the production of metabolites in newly eclosed adult mountain pines beetle *in situ* from late-fall to the coldest day of the winter in order to quantify their cold acclimation process.

## Materials and methods

Beetle collection began in late-summer (17 Sept 2016) at the Lucerne Campground (Robson Provincial Park, British Columbia, 52˚50'59.48"N, 118˚34'21.84"W, 1125m (Fig 1)). The infested stand consisted mostly of lodgepole pine (*Pinus contorta)* with evidence of recent mountain pine beetle attack dating from several previous years, including the most recent summer. Sampling continued weekly (25 Sept, 2 Oct, 11 Oct) and then bi-weekly (23 Oct, 6 Nov, 20 Nov, 3 Dec, 16 Dec) for nine total collection days during a 91-day period spanning almost the entire autumn of 2016. Brood galleries were exposed using a draw knife to remove bark from the tree. A minimum of 40 new adult beetles were collected from separate galleries in five randomly selected trees during each collection event (eight beetles per tree). New adults were identified based on their proximity to larval galleries and their relative size. For the first eight sampling days, beetles were confirmed to be living (by observing movement) prior to collection. Temperatures below –30˚C on the final day of sampling precluded this step as beetles were too cold for movement. Ice crystals were observed within sampled galleries on the final sampling day. Beetles were placed in either 2 mL or 1.5 mL dry snap-capped microcentrifuge tubes and flash frozen in the field using liquid nitrogen and then transported to the lab and

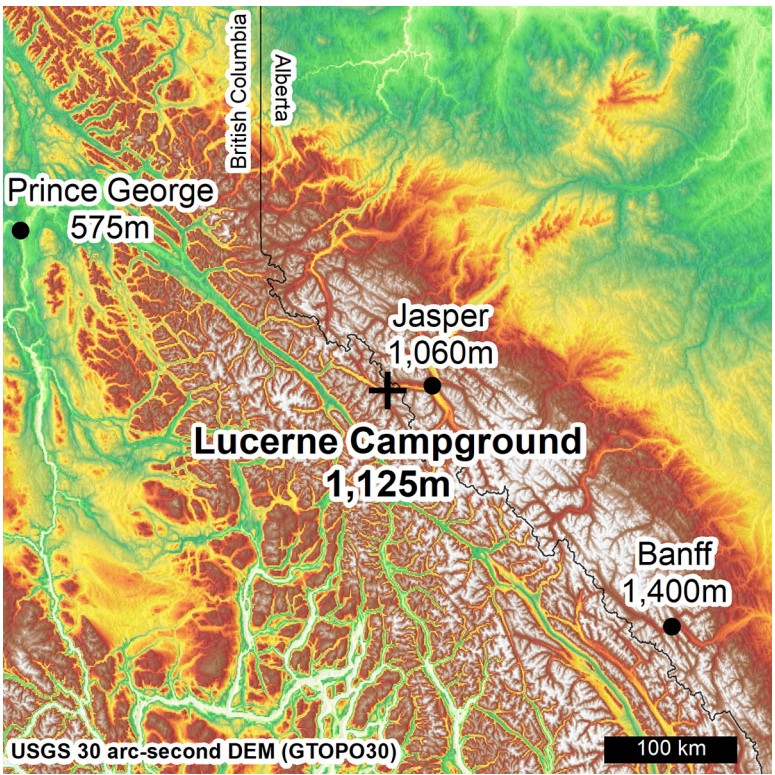

**Fig 1. Location of the study site (cross) in relation to the closest major urban centers (dots).** The sample site itself is located proximal to the Continental Divide of the Americas (latitude 52˚50'59.48"N, longitude 118˚34'21.84"W).

stored at −80˚C until metabolite processing. Three HOBO U23 Pro v2 data loggers (HOBO-ware, Onset Computer Corporation) were placed at breast height on well separated trees throughout the field site to track temperature. Climate loggers were moved from their original positions after the eighth sampling day and relocated to other trees near to their original positions within the sampling area due to sanitation logging removing the original trees used for placement.

Mountain pine beetles were quickly thawed on ice. Approximately 1g of beetle tissue was transferred to a mortar and pestle with 3 mL of chloroform:methanol (1:2 v/v). The beetles were then ground for 3 min and the extract was transferred to a 4 dram vial. The mortar was rinsed with 2 mL of a chloroform:methanol (1:2 v/v) mixture for the complete recovery of the extract. The entire extract was filtered using vacuum filtration. The residue was transferred into a 15 mL sterile screw-capped plastic centrifuge tube and 5 mL chloroform:methanol (1:2 v/v) mixture was added to the residue and was shaken at 250 rpm for 30 min at ambient temperature on a shaker. This extract was filtered again using vacuum filtration, combined with the first filtrate in a 13.5 mL Teflon lined screw-capped glass vial and the combined filtrate was transferred to a 50 mL sterile screw-capped plastic centrifuge tube. To this filtrate one quarter of the total volume of the filtrate 0.88% KCl was added. The tube was vortexed for 1.5 min and placed aside for 10 min for phase separation of an upper aqueous layer and lower organic layer. The tube was then centrifuged for 30 minutes at 3000 rpm. The upper aqueous layer (water-soluble metabolites) was transferred into a 15 mL sterile screw-capped plastic centrifuge tube and 2.5 mL HPLC water was added to the water-soluble metabolite extract and flash frozen in liquid nitrogen. This sterile screw-capped plastic centrifuge tube was lyophilized

with frozen water-soluble metabolites for 24 h and the resultant freeze-dried powder of was divided into 15 mg aliquots for NMR analysis.

A single 15 mg aliquot of the lyophilized water-soluble extract from pine beetles was taken in 1.5 mL snap-capped microcentrifuge tube. To this powder, 570 μL of water was added. The sample was sonicated for 15 min in a bath sonicator. To this sample, 60 μL of reconstitution buffer (585 mM phosphate buffer with 11.67 mM DSS) and 70 μL of D2O were added. The solution was vortexed for 1 min and centrifuged at 10,000 rpm for 15 min at ambient temperature. The clear supernatant was transferred into an NMR tube for NMR analysis.

All 1H-NMR spectra were collected on a 700 MHz Avance III (Bruker) spectrometer equipped with a 5 mm HCN Z-gradient pulsed-field gradient (PFG) cryoprobe. 1H-NMR spectra were acquired at 25°C using the first transient of the NOESY pre-saturation pulse sequence (noesy1dpr), chosen for its high degree of quantitative accuracy. All FID's (free induction decays) were zero-filled to 250 K data points. The singlet produced by the DSS methyl groups was used as an internal standard for chemical shift referencing (set to 0 ppm) and for quantification all 1H-NMR spectra were processed and analyzed using the online Bayesil software package. Bayesil allows for qualitative and quantitative analysis of an NMR spectrum by automatically and semi-automatically fitting spectral signatures from an internal database to the spectrum. Specifically, the spectral fitting for metabolites was completed using the standard serum metabolite library. Typically, all visible peaks were assigned. Most of the visible peaks are annotated with a compound name. This fitting procedure provides absolute concentration accuracy of 90% or better. Each spectrum was further inspected by an NMR spectroscopist to minimize compound misidentification and mis-quantification.

A one-way ANOVA was used to determine significance between measured metabolites, with a post-hoc Tukey's honestly significant different test used for pair-wise comparison of metabolite mean concentration between timepoints [adjusted p-value (FDR) = 0.05] following statistically significant (p < 0.05) ANOVA results. Both measures were conducted using MetaboAnalystR [24]. Concentration values and temperature measurements were visualized in Microsoft Excel (2016). Correlation of glycerol, proline, and trehalose to temperature values was performed in Rstudio (R version 3.4.4) using Pearson's product-moment correlation and negatively transformed temperature values.

## Results

Temperature at the study site initially decreased towards freezing but warmed and remained at or above freezing during the day during October and November. Temperatures dropped rapidly during early December and remained below –20°C for approximately two weeks, reaching the coldest on-site temperature of the winter (–30.1°C) on the final sampling day, 16 December 2016 (Fig 2). A spring collection was attempted when the site was again accessible following snow melt on 26 May 2017, but no living new adults could be found in a search of the area, including thorough investigation of trees previously sampled (number of trees checked on site > 50).

We detected 41 metabolites in overwintering adults (full list provided in S1 File). A one-way ANOVA with a post-hoc Tukey's HSD test for pairwise comparison showed 27 of these metabolites differed significantly at one or more of the time points taken during the study (Fig 3, but see also S2 File). Of these 27 significant measures, three metabolites–glycerol, trehalose, and proline–became highly elevated and are likely biologically significant in addition to being statistically significant (Fig 4). Glycerol concentrations were highest in beetles at the end of the sampling period, reaching 468.91 μg/mg of body weight (SE ± 49.74). Trehalose levels increased into the month of October but decreased in December, reaching a peak of

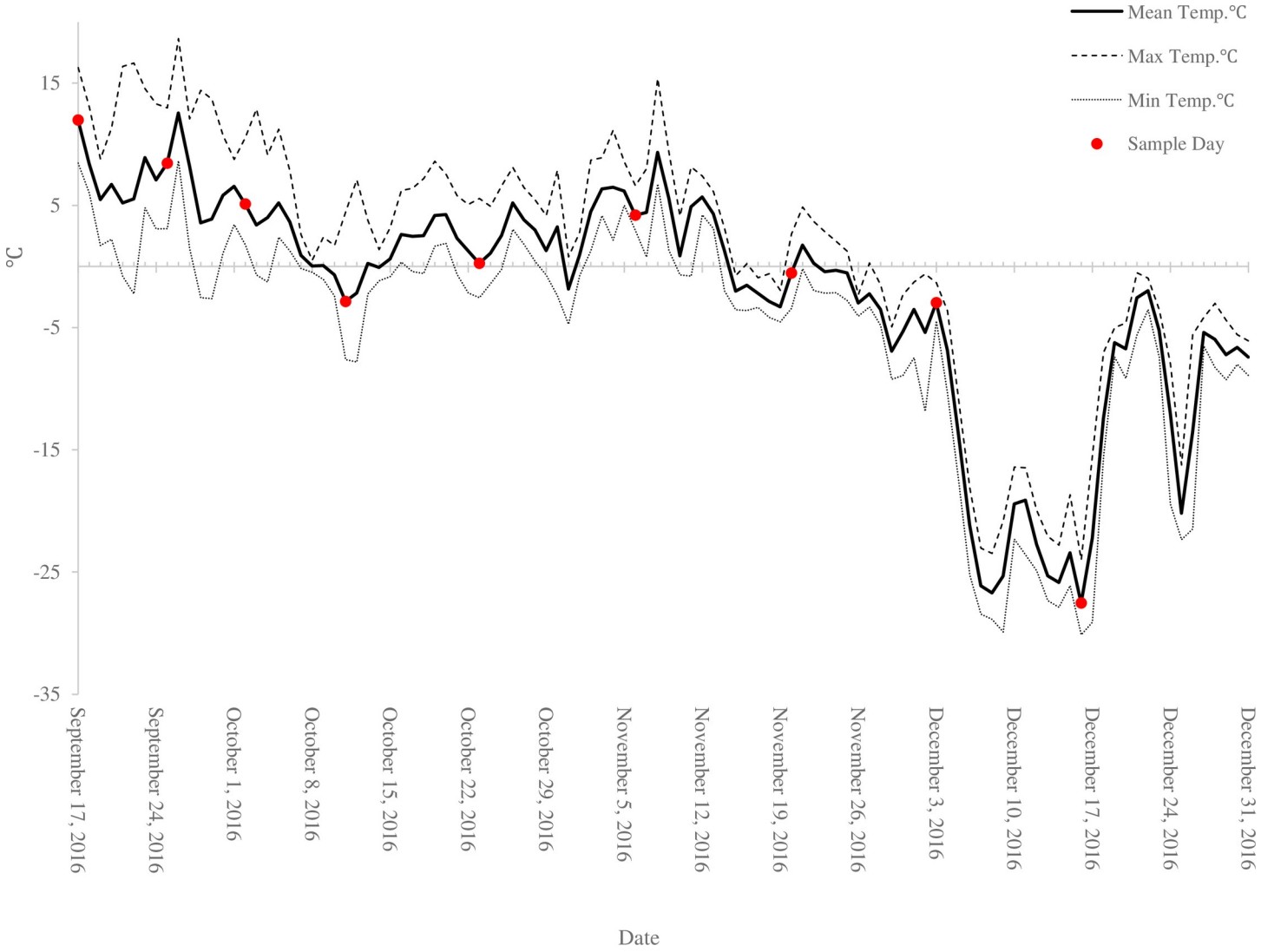

**Fig 2. Record of site ambient air temperature.** Local temperatures taken from climate data loggers at Lucerne Campground in Robson Provincial Park from September to December 2016.

180.16 µg/mg (SE ± 13.67) on 6 Nov. Proline levels increased until the middle of the sampling period and remained stable for the final four sampling days, reaching a final concentration of 46.02 µg/mg (SE ± 2.93) on the final sampling day in mid-December. Glycerol (p = 0.002) and proline (p = 0.016) both had a significant strong positive correlation to decreasing temperatures while trehalose had a non-significant correlation (Fig 5).

## Discussion

We found that new adult mountain pine beetles form their own metabolic antifreeze compounds in response to autumn temperature cues. The most responsive metabolites to increasing cold in adults were glycerol, trehalose, and proline, for all of which there is also previous transcriptomic and proteomic evidence of biosynthesis during larval cold hardening [15,16,22]. Mountain pine beetle larvae typically survive winters if ambient temperatures do not fall below approximately –40˚C, though larvae insulated below the snow line may survive

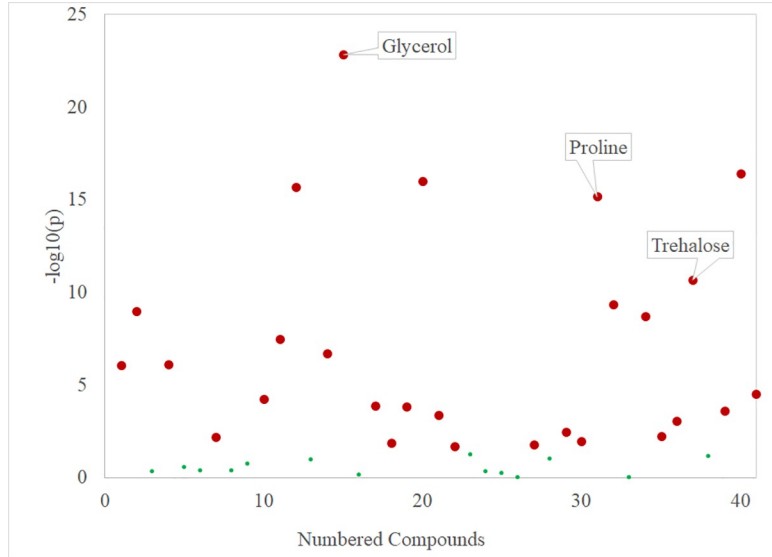

**Fig 3. One-way ANOVAs for all metabolites sampled.** All points marked in red exhibited significant differences between time points during the study while all points marked in green did not vary significantly during the study (FDR = 0.05, n = 8 samples per time point).

these cooler conditions [6,14]. Studies that have referenced the presence of newly eclosed adult mountain pine beetles and that tracked emergence rates have indicated that new adults have higher winter mortality rates compared to their larval counterparts, but did not investigate potential mechanisms for this reduced success [20,23]. We observed, but did not quantify, high mortality at our site during our site which we postulate is linked to several factors including site temperature regime, differing physiology compared to larvae, and below-bark conditions.

Glycerol is a known cryoprotectant in many insects, including other *Dendroctonus* spp., and has been documented in mountain pine beetle larvae [12,13,25,26]. It is a relatively inert compound that can be maintained at high concentration without interfering with other cellular processes or enzymatic reactions [27]. It is also nontoxic, so insects experience few fitness trade-offs when generating this compound, and it can be converted into glycogen when temperatures begin to warm [15,27,28]. Our previous studies have shown that larvae increase their capacity to generate glycerol in correlation to temperature similar to what we have observed here with adults [15,16,22] (Fig 4). Recent cold acclimation metabolite work has shown that mountain pine beetle larvae produce a concentration of glycerol an order of magnitude more per mg of tissue compared to the new adults profiled in our study (Batista pers. comm., visiting scholar, UNBC, batista@unbc.ca). This lower concentration of glycerol may have reduced the new adult beetles' ability to supercool and thereby increased mortality rates.

Trehalose is a major sugar constituent of insect haemolymph that acts as a mobile energy source for cellular respiration [27,29]. We observed increasing levels of trehalose in the mid- to late-fall, but no continual increase as temperatures grew cooler (Fig 4). In European populations of *Ips typographus* (Coleoptera: Curculionidae), trehalose undergoes a similar increase through October in response to cold temperatures [30]. Trehalose can also act as a cryoprotectant, stabilizing proteins at cold temperatures and keeping cellular membranes intact [31]. Increasing the durability of cellular membranes would reduce cellular damage in the event of changing osmotic pressure or ice crystal formation. Trehalose has also be linked to changing dietary cues for insects [29], meaning changing trehalose levels in the blood might lead to a

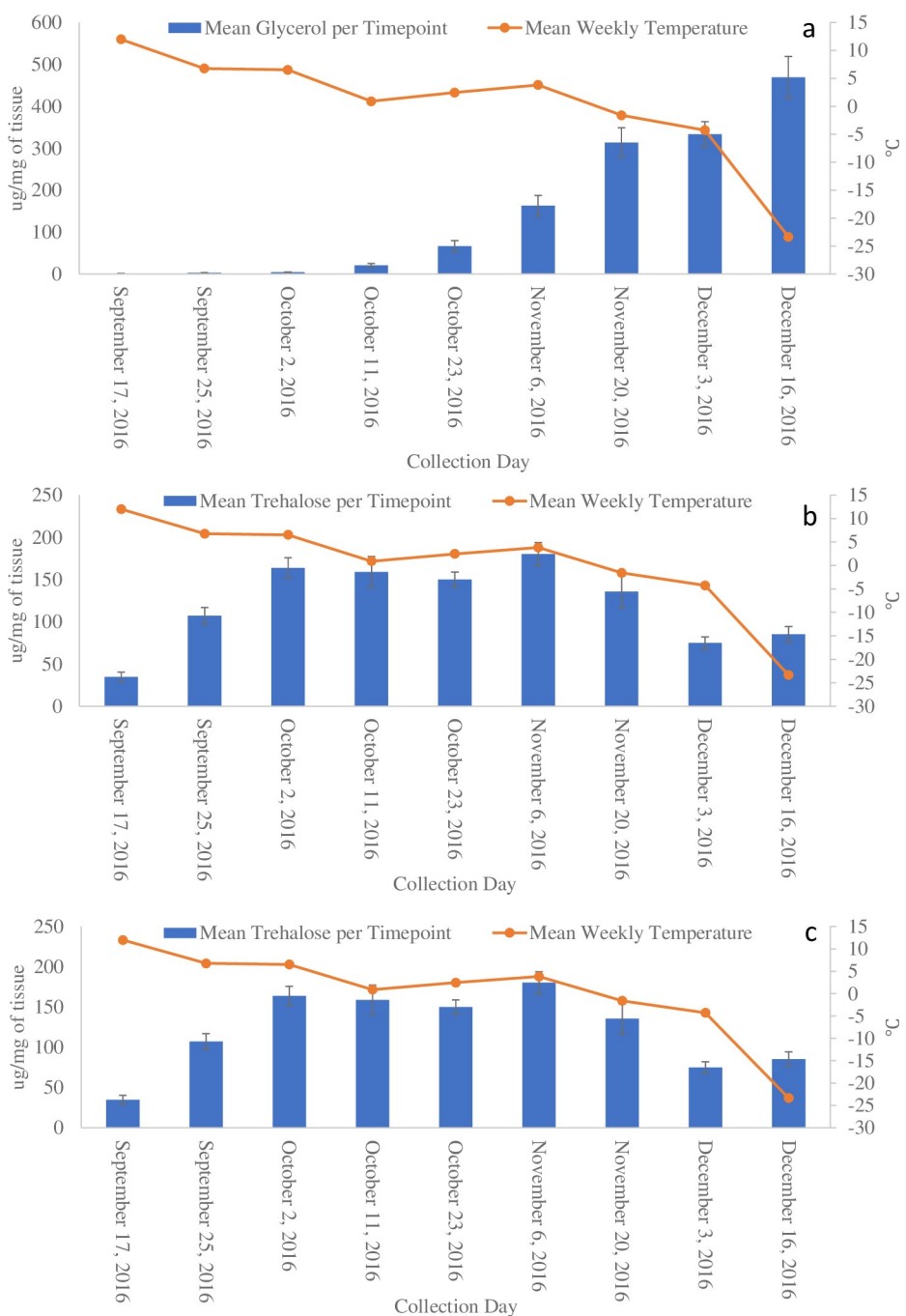

**Fig 4. Mean glycerol (a), trehalose (b), and proline (c) concentrations of new adult beetles (n = 8 samples per time point) in relation to ambient site temperature.** Error bars show standard error of the mean.

reduction or cessation of feeding behavior as temperatures cool, helping with voiding of the gut prior to onset of winter.

Proline is known to be a cryoprotectant in both plant and yeast cells [32] but is not well-documented as a cryoprotectant in insects. The increase of proline levels in response to temperature within the new adults suggests a connection to cold acclimation in mountain pine beetles (Fig 4). In the red flat bark beetle, *Cujucus clavipes* (Coleoptera: Cucujidae), proline

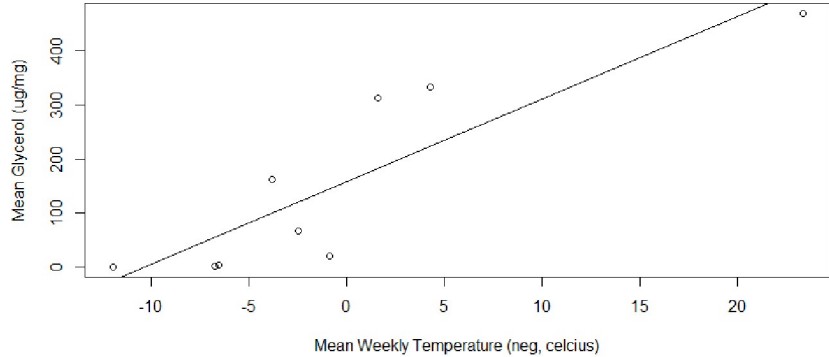

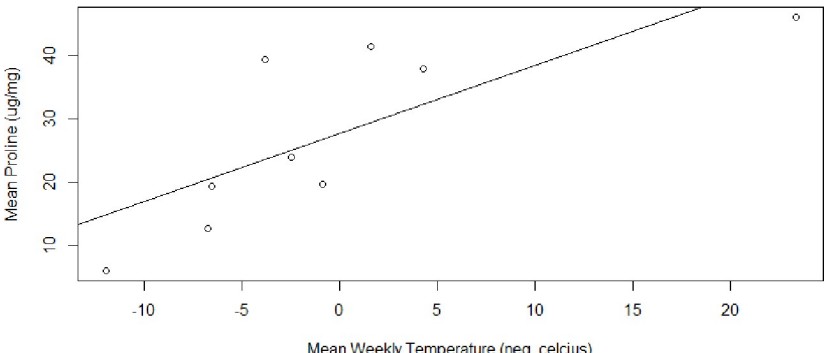

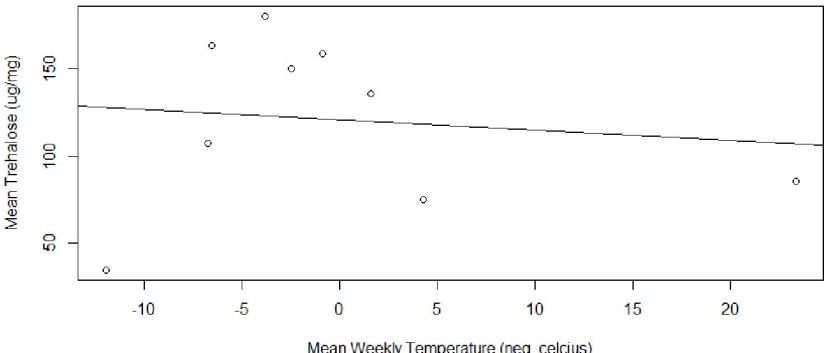

**Fig 5. Pearson's product-moment correlation between mean metabolite concentration and negatively transformed temperature (˚C) over time (n = 8 samples per timepoint, μg/mg of tissue).** Mean glycerol concentration (r = 0.873, p < 0.01, R2 = 0.7615) and mean proline concentration (r = 0.767, p < 0.05, R2 = 0.588) were both significant while mean trehalose concentration (r = 0.125, p > 0.05, R2 = 0.0157) was non-significant.

and alanine are thought to work together with trehalose to slow the freezing process [33]. Alanine was detected in the new adults, though it did not correlate with temperature over the duration of the study (S1 File). It is possible that new mountain pine beetle adults use proline to decrease their supercooling point, though further study would be needed to confirm this possibility. Proline is also metabolized along with carbohydrates during flight [34,35]. Proline generation may serve a dual purpose where beetles use the amino acid as a cryoprotectant in the winter and then metabolize remaining proline as a flight fuel for dispersal.

On-site phenology drivers and additional metabolic demands on new adults are likely contributing factors to the level of mortality observed in the field. While field temperatures initially dropped at a steady rate from September to early October, they rewarmed between mid-October and November (Fig 3). The extended period of warmer weather may have confused the cues that normally trigger the beetles to produce cryoprotectants. New adults feed on fungal associates after eclosing but prior to emergence from under the bark [6]; based on our observations of bark and phloem conditions, adequate food resources were available to beetles on the study site. Adults beetles do, however, have different metabolic demands compared to larvae. They must develop fat reserves to support flight and also maintain gonadal tissue for reproduction [10,35]. It should be noted that adult females partition their metabolic resources to at least some extent–for instance, they do not generate vitellogenin until they come into contact with a host tree following dispersal flight [36]. Larvae have not yet developed these tissues beyond imaginal disks, and may thus have more resources to allocate to cold hardiness.

As ice crystals were observed within galleries on the coldest sampling day, this may have been a contributing factor to the increased mortality observed on the sampling site. Direct contact of ice crystals on the surface of an insect's exoskeleton creates a surface where point nucleation of ice can occur [14,37]. New adults are melanized with hardened carapaces and have more surface area compared to larva. Having undergone pupation, new adults also have thin legs that are liable to freeze faster due to their exposure. It is probable that beetles with elevated levels of cryoprotectants still experienced internal ice crystal formation due to the external ice contact. In addition, it is unknown if new adults are capable of voiding their guts as larvae do in preparation for freezing temperatures [18]. If they are not capable, new adults would have more internal surfaces for ice crystal formation due to retained food, likely making them yet more susceptible to freezing.

We found overwintering, newly eclosed mountain pine beetle adults produce three known antifreeze metabolites. Previous research suggests that these are the same three major metabolites produced by larvae during cold acclimation [15,16,22], but it is likely newly eclosed adults produce less of each cryoprotectant. This is the first time that a metabolic mechanism for new adult survival has been documented. While the new adults in our study experienced high mortality, larvae at nearby sites in Jasper National Park were found to have survived the winter of 2016–2017. New adult mountain pine beetles have been most commonly described at high elevation, due to a comparatively late start to spring and early start to cooling autumn temperatures [20,38]. In areas where winters begin earlier in the year, thus reducing the cold acclimation period, and have temperatures reaching below –30˚C for several days, it is likely that new adult beetles that experience an extended life cycle will not be able successfully overwinter. Our observation of high mortality further supports the original field records of lower overwintering success in new adults [20,23], and suggests both metabolic and physical drivers. This work provides new parameters for modeling the spread of mountain pine beetles in their expanding geographic range and is evidence of physiological plasticity in this insect. In both novel, colder regions like the Boreal Forest and warmer ecosystems which experience more developmental degree days, we may see the effect of natural selection amplifying varied life cycle lengths (longer or shorter) in mountain pine beetle.

## Supporting information

**S1 File. Mean metabolite concentrations by temperature.** An excel sheet containing data for all metabolites in both tabular and graphical form.
(XLSX)

**S2 File. ANOVA data and expanded Fig 3.** An excel sheet containing all ANOVA output data and an annotated version of Fig 3 with all components named.
(XLSX)

## Acknowledgments

We are grateful to the British Columbia Ministry of Environment and Climate Change Strategy (Park Use Permit, Authorization # 107171) and Robson Provincial Park staff for providing on-site assistance. Thanks also to Dr. Lisa Poirier for providing the data loggers used in this study. Thanks to my husband, Neil Thompson, for calling me back from Alberta so we could sample on the coldest day of the year, and to Daisy, the bravest little canine field assistant ever.

## Author Contributions

**Conceptualization:** Kirsten M. Thompson, Dezene P. W. Huber, Brent W. Murray.

**Data curation:** Kirsten M. Thompson.

**Formal analysis:** Kirsten M. Thompson.

**Funding acquisition:** Kirsten M. Thompson, Dezene P. W. Huber, Brent W. Murray.

**Investigation:** Kirsten M. Thompson.

**Methodology:** Kirsten M. Thompson, Dezene P. W. Huber.

**Project administration:** Kirsten M. Thompson.

**Resources:** Kirsten M. Thompson.

**Software:** Kirsten M. Thompson.

**Supervision:** Kirsten M. Thompson, Dezene P. W. Huber, Brent W. Murray.

**Validation:** Kirsten M. Thompson.

**Visualization:** Kirsten M. Thompson.

**Writing – original draft:** Kirsten M. Thompson.

**Writing – review & editing:** Kirsten M. Thompson, Dezene P. W. Huber, Brent W. Murray.

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
