## [Decision Letter · Decision Letter 0]

13 Aug 2019

PONE-D-19-16184

Autumn shifts in cold tolerance metabolites in overwintering adult mountain pine beetles

PLOS ONE

Dear Mrs. Thompson,

Thank you for submitting your manuscript to PLOS ONE. After careful consideration, we feel that it has merit but does not fully meet PLOS ONE’s publication criteria as it currently stands. Therefore, we invite you to submit a revised version of the manuscript that addresses the points raised by the reviewers. While both reviewers' comments do not require substantial revisions, they do raise a number of points that are important for improving your manuscript.

We would appreciate receiving your revised manuscript by Sep 26 2019 11:59PM. To enhance the reproducibility of your results, we recommend that if applicable you deposit your laboratory protocols in protocols.io, where a protocol can be assigned its own identifier (DOI) such that it can be cited independently in the future. For instructions see: http://journals.plos.org/plosone/s/submission-guidelines#loc-laboratory-protocols

We look forward to receiving your revised manuscript.

Kind regards,

Christopher Bone

Academic Editor

PLOS ONE

Journal Requirements:

1. Please amend your list of authors on the manuscript to ensure that each author is linked to an affiliation. Authors’ affiliations should reflect the institution where the work was done (if authors moved subsequently, you can also list the new affiliation stating “current affiliation:….” as necessary).

Reviewers' comments:

Reviewer's Responses to Questions

**Comments to the Author**

1. Is the manuscript technically sound, and do the data support the conclusions?

Reviewer #1: Partly

Reviewer #2: Yes

2. Has the statistical analysis been performed appropriately and rigorously? 

Reviewer #1: Yes

Reviewer #2: Yes

3. Have the authors made all data underlying the findings in their manuscript fully available?

Reviewer #1: Yes

Reviewer #2: Yes

4. Is the manuscript presented in an intelligible fashion and written in standard English?

Reviewer #1: Yes

Reviewer #2: Yes

5. Review Comments to the Author

Reviewer #1: This manuscript provides novel information concerning the acclimation process of metabolites in two-year mountain pine beetle. I recommend accept following moderate revisions and offer the following suggestions for improvement:

1) Ln. 24, 112: Revise to “…nuclear magnetic resonance (NMR).”

2) Ln. 26: Revise to “…beetles appear to prepare…”

3) Ln. 27-28: This statement is not substantiated by the data as concentrations of metabolites were not analyzed for larvae. Review and revise as appropriate throughout the ms.

4) Ln. 73: Replace “campground” with “Campground”

5) Ln. 75: Delete “is located very near to the Great Divide”; redundant w/ ln. 93.

6) Ln. 81: Define how it was determined these were “new adult beetles” vs. re-emergent adults.

7) Ln. 95: Revise to “Mountain pine beetle larvae…”

8) Ln. 144: Was this the “coldest temperature of the winter” (12/21–3/20) or during the collection period? Review and revise as appropriate.

9) Ln. 179-180: Don’t these statements largely reflect the same condition (i.e., snow insulates within bark temperatures)? Review and revise as appropriate.

10) Ln. 196: Provide complete contact information for Batista.

11) Ln. 231: Revise to “…they do not…”

12) Ln. 234: Revise to “As ice crystals were observed within galleries on the coldest sampling day, this may have been…”

13) Ln. 243: Revise to “…likely making…”

14) Ln. 245-246: Revise to “…mountain pine beetle adults produce three…”

15) Ln. 250: Replace “park” with “Park”

16) Ln. 251 (and throughout): Delete “MPB” or introduce at first mention of mountain pine beetle and carry throughout the ms.

17) Ln. 253-255: Delete sentence.

18) Ln. 255: Does the length of the winter matter? Revise.

19) Ln. 256: Revise to “…-30 ℃ for several days…”

20) Ln. 260: Do you expect similar results in jack pine? Elaborate.

21) Figure 2: Add dates to X-axis.

Reviewer #2: Review of PLOS-Pne, D-19-16184 by Thompson et al.

This is a well-written paper presenting important new information on the overwintering ecology of mountain pine beetle. The overwintering capability of adult beetles has been scantly examined.

The paper is worthy of publication because of the new findings of cryoprotectans in adult beetles.

Although I am recommending a minor revision there are two items that the authors should address.

First, I am always hesitant to accept findings from a field study of this type with only one year of data. A minimum of two years at least allows some level or repeatability. Environmental factors, specifically in this case temperature, can influence findings in unpredictable ways. The authors acknowledge the possible influence in lines 222-225 where they indicate “While field temperatures initially dropped at a steady rate from September to early October, they rewarmed between mid-October and November (Fig 3). The extended period of warmer weather may have confused the cues that normally trigger the beetles to produce cryoprotectants.” I would offer that the authors consider offering some justification and support as to why data from only one year can be considered reliable.

Second, throughout the discussion the authors refer to “increased mortality rates” (of beetles) or statements along these lines. Mortality was not quantified (as far as I can tell). The statement seems to refer to the observation of no live beetles in their last sampling date. To infer increased mortality rates by examining 5 trees in a single date does not seem appropriate. Within a small area, even in the same stand it is possible to find a few trees without surviving beetles and trees with surviving beetles. I would recommend that the authors remove this from the discussion.

Also important – on Line 95 in the methods, the paper talks about larvae being processed. Is this correct? Or is this supposed to be adults?

Line 35, suggest remove the comment of “unprecedented” as other studies are showing that this is unlikely the case.

Line 39, although this is well established, think it is always good to have a link to a weather data website of a citation that provides data.

Line 47, are the beetles really freeze intolerant? They have evolved a biological mechanism to withstand freezing temperatures.

Line 37, are these air or subcortical temperatures? Likely do not differ too much but would be good to clarify.

Lines 90-91, this sentence is confusing, please clarify.

Line 112, please spell out NMR so the unfamiliar reader can understand or at least search what the technique is.

Figure 2, again – air temperatures?

Line 153, from Figure 3, seems like 3 compounds were elevated as much as Proline – are these not important? Should these be discussed? What are they?

Figure 4, may be better to have the actual sampling dates in the x-axis.

Line 160 and Figure 5, not sure I follow this – negative correlations but the line is increasing. The x-axis goes from -10 to 20 C but the x-axis label indicates neg Celsius. Are the numbers on the x-axis incorrect?

This is a nice and relevant contribution to our understanding of mountain pine beetle biology. With some minor clarifications this will be a very good straightforward and concise paper delivering important findings.

Sincerely,

José F. Negrón

Rocky Mountain Research Station

6. PLOS authors have the option to publish the peer review history of their article (what does this mean?). If published, this will include your full peer review and any attached files.

Reviewer #1: No

Reviewer #2: Yes: Jose F. Negron

---

## [Author Response · Author response to Decision Letter 0]

22 Sep 2019

We are grateful to both reviewers for their helpful suggestions and comments. We also thank the editor for his suggestions on document formatting and style requirements. All revisions are detailed in the response to reviewers document that has been uploaded to the editorial manager.

---

## [Editor Report · Decision Letter 1]

16 Dec 2019

Autumn shifts in cold tolerance metabolites in overwintering adult mountain pine beetles

PONE-D-19-16184R1

Dear Dr. Thompson,

We are pleased to inform you that your manuscript has been judged scientifically suitable for publication and will be formally accepted for publication once it complies with all outstanding technical requirements.

With kind regards,

Christopher Bone

Academic Editor

PLOS ONE

Reviewers' comments:

Thank you for providing a thorough response to the reviewers' comments. You have addressed all existing concerns, and as such no further revisions are being requested.

---

## [Editor Report · Acceptance letter]

23 Dec 2019

PONE-D-19-16184R1 

Autumn shifts in cold tolerance metabolites in overwintering adult mountain pine beetles 

Dear Dr. Thompson:

I am pleased to inform you that your manuscript has been deemed suitable for publication in PLOS ONE. Congratulations! Your manuscript is now with our production department. 

With kind regards,

on behalf of

Dr. Christopher Bone 

%CORR_ED_EDITOR_ROLE%

PLOS ONE